# Constructing phase boundary in AgNbO$_3$ antiferroelectrics: pathway simultaneously achieving high energy density and efficiency

Nengneng Luo [1,2,8✉], Kai Han[1,8], Matthew J. Cabral[3,8], Xiaozhou Liao [3], Shujun Zhang [4✉], Changzhong Liao[5], Guangzu Zhang[6], Xiyong Chen[1], Qin Feng[1], Jing-Feng Li [7] & Yuezhou Wei[1✉]

Dielectric capacitors with high energy storage density ($W_{rec}$) and efficiency ($\eta$) are in great demand for high/pulsed power electronic systems, but the state-of-the-art lead-free dielectric materials are facing the challenge of increasing one parameter at the cost of the other. Herein, we report that high $W_{rec}$ of 6.3 J cm$^{-3}$ with $\eta$ of 90% can be simultaneously achieved by constructing a room temperature M2–M3 phase boundary in (1-$x$)AgNbO$_3$-$x$AgTaO$_3$ solid solution system. The designed material exhibits high energy storage stability over a wide temperature range of 20–150 °C and excellent cycling reliability up to 10$^6$ cycles. All these merits achieved in the studied solid solution are attributed to the unique relaxor antiferroelectric features relevant to the local structure heterogeneity and antiferroelectric ordering, being confirmed by scanning transmission electron microscopy and synchrotron X-ray diffraction. This work provides a good paradigm for developing new lead-free dielectrics for high-power energy storage applications.

[1] Guangxi Key Laboratory of Processing for Non-ferrous Metallic and Featured Materials, School of Resources, Environment and Materials, Guangxi University, 530004 Nanning, China. [2] Center on Nanoenergy Research, School of Physical Science and Technology, Guangxi University, 530004 Nanning, China. [3] School of Aerospace, Mechanical & Mechatronic Engineering, The University of Sydney, Sydney, NSW 2006, Australia. [4] Institute for Superconducting and Electronic Materials, Australian Institute of Innovative Materials, University of Wollongong, Wollongong, NSW 2500, Australia. [5] Department of Civil Engineering, The University of Hong Kong, Pokfulam Road, Hong Kong, SAR, China. [6] School of Optical and Electronic Information, Huazhong University of Science and Technology, 430074 Wuhan, China. [7] State Key Laboratory of New Ceramics and Fine Processing, School of Materials Science and Engineering, Tsinghua University, Beijing, China. [8] These authors contributed equally: Nengneng Luo, Kai Han, Matthew J. Cabral. ✉email: luonn1234@163.com; shujun@uow.edu.au; yzwei@gxu.edu.cn

Dielectric capacitors are widely utilized in numerous advanced high/pulsed power electronic systems, due to their distinctive features of high power density, ultrafast charge/discharge capability, long storage lifetime, robust, and excellent thermal stability[1–4]. However, they possess inferior energy density in comparison with other electrochemical energy storage systems such as batteries. Therefore, extensive efforts have been made to improve their energy densities to meet the demands of integration, compactness, and miniaturization of electronic devices[5,6]. In addition, from practical application viewpoint, high energy efficiency ($\eta$) is desired since the energy dissipation will greatly degrade the thermal breakdown strength thus impact the reliability and performance of the energy storage capacitors. However, previous investigations have shown that the energy density and efficiency can be enhanced only at the expense of each other for most dielectric materials.

Dielectric materials developed for energy storage capacitors include linear dielectrics (LD), ferroelectrics (FEs), antiferroelectrics (AFEs), and relaxor ferroelectrics (RFEs)[5]. Among them, AFEs have been attracted extensive attention for energy storage application because of their unique double hysteresis loop originating from the electric field induced antiferroelectric-ferroelectric (AFE–FE) phase transition and zero remnant polarization ($P_r$) in pristine AFE phase. These advantages have been fully reflected in PbZrO$_3$-based AFE ceramics, in which large energy storage densities ranging from 6.4 to 11.2 J cm$^{-3}$ were reported[7–9]. However, the disadvantage of AFE is high energy loss due to large hysteresis associated with the first-order AFE–FE phase transition, being confirmed by its polarization vs. electric field ($P–E$) loop[10]. It is thus a challenge to achieve high energy storage density and efficiency simultaneously in antiferroelectric materials.

On the other hand, RFEs exhibit hysteresis-free polarization response owing to the existence of local structure heterogeneity, thus leading to a high energy efficiency[11–14]. Analogous to RFEs, it is expected that relaxor antiferroelectrics (RAFEs) might be a good choice to address the hysteresis, which inevitably exists in AFE, where disruption of the long-range ordered AFE domains will smear the AFE–FE phase transition due to the weakly intercoupled nanodomains[15,16]. Based on this concept, a paraelectric or relaxor ferroelectric end member was judiciously introduced to AFEs to break the long-range AFE order into nanodomains. The introduction of $(Sr_{0.7}Bi_{0.2})TiO_3$ relaxor end member into $(Na_{0.5}Bi_{0.5})TiO_3$ forms a new RAFEs solid solution with high energy efficiency of 95% and energy storage density of 2.5 J cm$^{-3}$[16]. In addition, an ultrahigh energy storage density of 12.2 J cm$^{-3}$ was achieved in $(Bi_{0.5}Na_{0.5})TiO_3$-NaNbO$_3$ ceramics, where the relaxor component $(Bi_{0.5}Na_{0.5})TiO_3$ was added into NaNbO$_3$, with the purpose to stabilize the room temperature antiferroelectric phase in NaNbO$_3$ and introduce relaxor feature[17], but with less success in energy efficiency being below 70% due to the inferior AFE stability.

To achieve both high energy storage density and efficiency simultaneously, we propose to design material system with a highly stabilized antiferroelectricity with relaxor feature. AgNbO$_3$ (AN) has been actively studied for dielectric energy storage application, due to its unique antiferroelectric feature. It undergoes a series of phase transitions with increasing temperature, possessing a ferrielectric (FIE) M1 phase and two disordered AFE phases (M2 and M3) below Curie temperature[18,19], as shown in Fig. 1a. At room temperature (RT), the M1 phase exhibits metastable AFE feature under applied electric field, leading to a large $P_r$ and hysteresis in the $P–E$ loops, as shown in Fig. 1b. Nevertheless, a relatively high $W_{rec}$ of 1.5–2.0 J cm$^{-3}$ was obtained in AN, with an efficiency only around 38%[20,21]. Numerous attempts have been made on A-site[22–24], B-site[25,26],

and A/B-site[27,28] chemical modifications with the idea to shift the highly stable AFE M2 phase to RT, where $P–E$ loop with small $P_r$ and reduced hysteresis have been achieved (Fig. 1c), exhibiting good energy storage densities varying in the range of 2.5–4.5 J cm$^{-3}$ and efficiency of 55–69%. A more attractive energy storage density of 5.2 J cm$^{-3}$ was reported in $Ag_{0.91}Sm_{0.03}NbO_3$ ceramics but with yet low efficiency of 68.5%[29]. It should be noted that the disorder feature can be induced by aliovalent ion dopant in AN system, which can be confirmed by the obvious frequency dispersion over M1–M2 phase transition and high diffuseness parameter[30]. This may be associated with the downward shifting of M2–M3 phase transition temperature with diffused dielectric maximum[31], being generally ascribed to the different degrees of displacement orders in M2 and M3 phases[32]. Therefore, relaxor characteristic with slim $P–E$ loop is expected if M2–M3 phase transition temperature ($T_{M2-M3}$) shifts downward to RT, which can be represented by the minimized $P_r$ and hysteresis as given in Fig. 1d, leading to high energy storage density and efficiency simultaneously.

Analogous to AN, the AgTaO$_3$ (AT) material possesses numerous phase transitions as a function of temperature, with room temperature rhombohedral phase[18,33]. Of particular significance is that AN and AT have the infinite miscibility thus (1-x)AgNbO$_3$-xAgTaO$_3$ (ANTx) solid solution can be formed with the addition of AgTaO$_3$ in AgNbO$_3$, leading to successive physical property transformations, such as phase transition sequence and dielectric permittivity depending on solid solution composition[34]. It is expected that the M1–M2 and M2–M3 phase transition temperatures of AN can be tailored over a wide temperature range downward to below RT with addition of AT, which provides a feasible way for tuning the physical properties of the solid solution.

In this work, we designed the ANTx solid solutions, where the M2–M3 phase boundary was built at RT with significantly stabilized antiferroelectric phase, meanwhile possessing relaxor features. As expected, high energy storage density of 6.3 J cm$^{-3}$ and efficiency of 90% were achieved simultaneously. In addition, the local structure heterogeneity and antiferroelectric ordering of M2–M3 phase boundary were confirmed on the atomic scale, giving a solid proof on the long-term confusion about the broad dielectric anomaly of $T_{M2-M3}$.

## Results

**Dielectric property of ANTx.** Figure 2a gives the temperature-dependent dielectric permittivity and loss for the representative ANTx compositions. Typical dielectric anomalies associated with various phase transitions are observed in AN, consistent with the results reported previously[20]. Similar phase transition behaviours are observed for ANTx solid solutions. As expected, the phase transition temperatures $T_{M1-M2}$ and $T_{M2-M3}$ are found to shift significantly downward. Of particular importance is that the $T_{M2-M3}$ of ANT55 composition decreases to around RT, as represented in the schematic phase diagram of Fig. 2b. In addition, the phase transition of $T_{M1-M2}$ is found to become smeared and diffused with clear frequency dispersion over a broad temperature range, as shown in Fig. 2c, indicating a typical relaxor feature[35]. To further analyze the relaxor behaviour of ANTx ceramics, the frequency dispersion is calculated based on $\triangle T = T_{M1-M2}$ (100 kHz) $- T_{M1-M2}$ (100 Hz), where the $T_{M1-M2}$ (100 kHz) and $T_{M1-M2}$ (100 Hz) are determined by the dielectric anomalies. The $\triangle T$ is found to increase with increasing Ta content, indicating the addition of Ta component in AN will induce a strong relaxor characteristics, which will greatly benefit the energy storage density and efficiency.

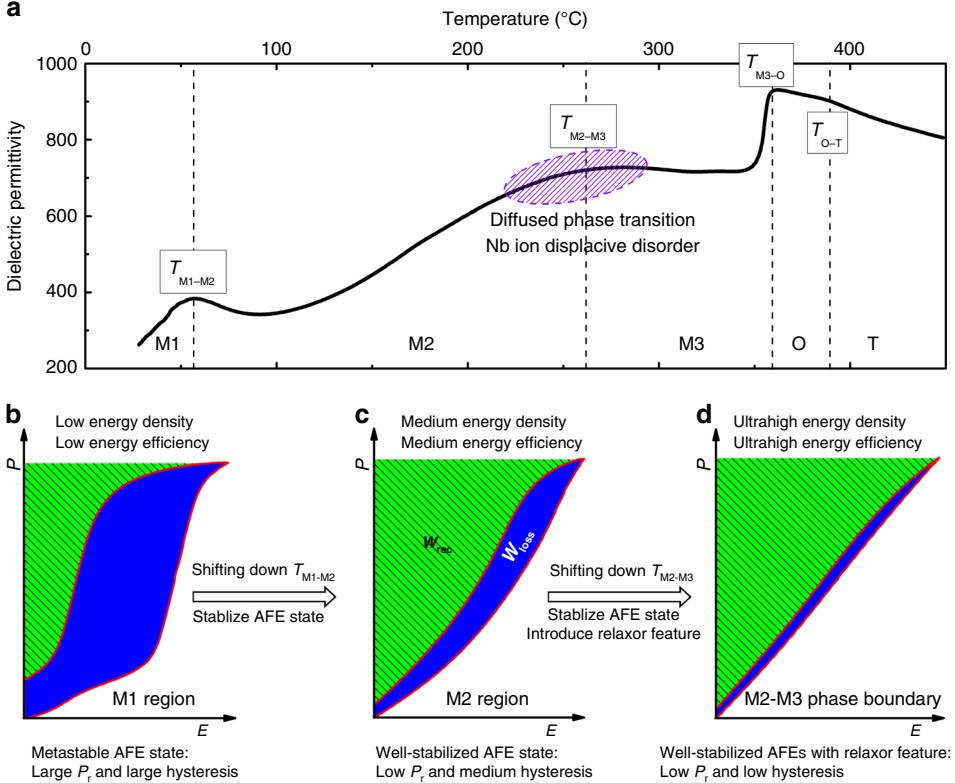

**Fig. 1 Designing principles of high energy storage performance AN-based materials. a** Temperature dependence of the dielectric permittivity of AN with a series of phase transitions. The $T_{M1-M2}$ stands for M1–M2 phase transition temperature. Schematic diagram of P–E loops for energy storage of **b** a metastable AFEs with phase in M1 region, **c** fully stabilized AFEs with phase in M2 region, **d** fully stabilized AFEs with relaxor characteristics around M2–M3 phase boundary where a diffused phase transition can be observed.

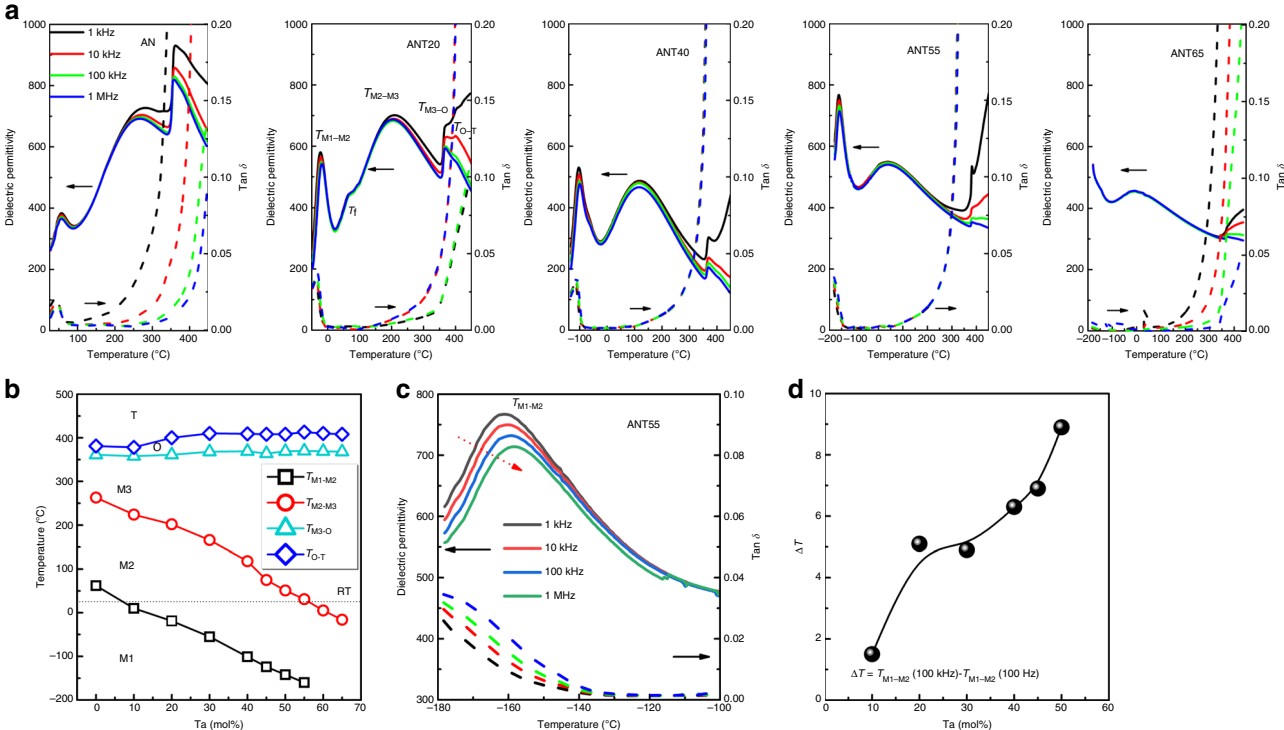

**Fig. 2 Dielectric properties of ANTx ceramics. a** Temperature- and frequency-dependent dielectric permittivity and loss. **b** Schematic phase diagram based on the temperature-dependent dielectric permittivity. **c** Dielectric permittivity and loss of ANT55 ceramic over the temperature range from −180 to −100 °C. **d** Composition dependence of the frequency dispersion △T.

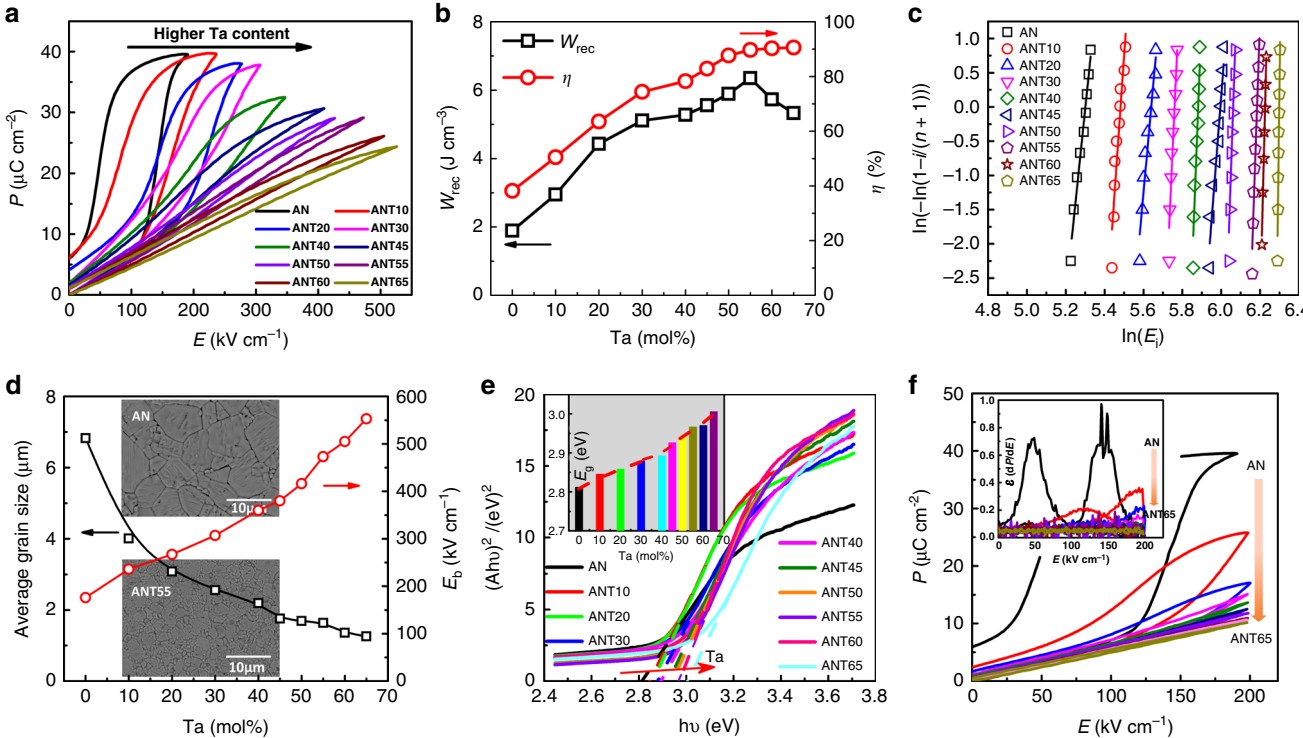

**Fig. 3 Ferroelectric and energy storage performance of ANTx ceramics, and their breakdown analysis. a** The $P–E$ loops. **b** Composition-dependent energy storage properties ($W_{rec}$ and $\eta$). **c** Weibull distribution of the breakdown electric field $E_b$ on samples with thickness of ~0.15 mm. **d** The composition dependence of average grain size and $E_b$, the insets give the SEM micrographs of the AN and ANT55 ceramics, respectively. **e** UV–vis absorption spectra of AN and ANTx. The inset is the composition-dependent band gap $E_g$. **f** The composition-dependent polarizations (AN:190 kV cm$^{-1}$; others:200 kV cm$^{-1}$), the inset gives the field dependent dielectric permittivity ($\varepsilon = dP/dE$).

**Energy storage performance of ANTx.** To evaluate the energy storage performance of the as-designed ceramics, $P–E$ loops are measured prior to their corresponding breakdown strengths at a frequency of 1 Hz, as shown in Fig. 3a. The $P–E$ loops conform to the typical feature of AFEs, where the AFE–FE phase transition is shifted to higher electric fields with the addition of Ta. Of particular importance is that the remnant polarization monotonically decreases with the increase of Ta content, reaching the value of zero at $x = 65$ mol%, while the hysteresis of $P–E$ loops reduces obviously with nearly hysteresis-free feature at compositions with $x$ above 50 mol%. The obvious evolutions of AFE–FE phase transition electric field and remnant polarization, together with the less electric field dependent dielectric permittivity with increasing Ta concentration (Supplementary Fig. 1), give solid proof of high stability of antiferroelectricity in ANT system. It should be noted here that the relaxor characteristic would smear the $P–E$ loops, which also increases AFE–FE phase transition electric field and decreases remnant polarization. As a consequence, the energy storage efficiency $\eta$ is remarkably increased with addition of Ta, reaching above 90% for ANTx with $x > 55\%$ (Fig. 3b). Of particular significance is that ultrahigh energy storage density up to 6.3 J cm$^{-3}$ is achieved for ANT55, showing a pronounced enhancement of ~330% comparing to 1.9 J cm$^{-3}$ for the pure AN counterpart. The ultrahigh energy storage density is closely associated with a high breakdown strength ($E_b$). Figure 3c gives the $E_b$ values based on the Weibull distribution, in which good linear relationship between X and Y axes can be observed for all compositions. The $E_b$ increases substantially after the addition of Ta, demonstrating ultrahigh values of 470 and 550 kV cm$^{-1}$ for ANT55 and ANT65 (Fig. 3d), respectively. To understand the underlying mechanisms responsible for the significantly improved $E_b$ in ANTx solid solution, the micro-morphology and

grain size, the band gap, as well as the polarizations at high electric fields are studied. All ANTx ceramics show highly compacted grains with nearly pore-free microstructure (insets of Fig. 3d), leading to high relative bulk density of >96%. The average grain size decreases obviously with the increase of Ta content (6.8 μm for AN vs. 1.3 μm for ANT65), as represented in Fig. 3d, due to the refractory nature of Ta$_2$O$_5$[36]. More detailed composition-dependent micro-morphology and average grain size distribution can be found in Supplementary Figs. 2 and 3, respectively. The high relative bulk density and reduced grain size will greatly benefit the enhanced breakdown strength[37]. Furthermore, improved band gap ($E_g$) is observed with the increase of Ta content, with values ranging from ~2.8 eV (AN) to ~3.0 eV (ANT65), obtained from the UV–vis absorption spectra (Fig. 3e). The wider band gap will make the electrons in the valence band more difficult to jump into the conduction band, which contributes to a higher intrinsic breakdown strength[38]. Finally, the polarization is clearly decreased after Ta addition, as shown in Fig. 3f, which is associated with the smeared AFE–FE phase transition and relaxor behaviour. The decreased polarization results in much lower dielectric permittivity (dP/dE) maxima upon AFE–FE phase transition, as shown in the inset of Fig. 3f. A moderate increase in polarization and/or dielectric permittivity as a function of applied electric field will impede the dramatic enhancement in electric energy density, thus leading to improved breakdown strength[39]. All the above-mentioned factors are responsible for the significantly increased breakdown strength for ANTx ceramics.

To verify the antiferroelectric characteristics of ANT55, the electric field dependent $P–E$ loops and current vs. voltage ($I–V$) curves were measured, as given in Fig. 4a, b, respectively. Linear $P–E$ loops with very low polarization are observed at low electric

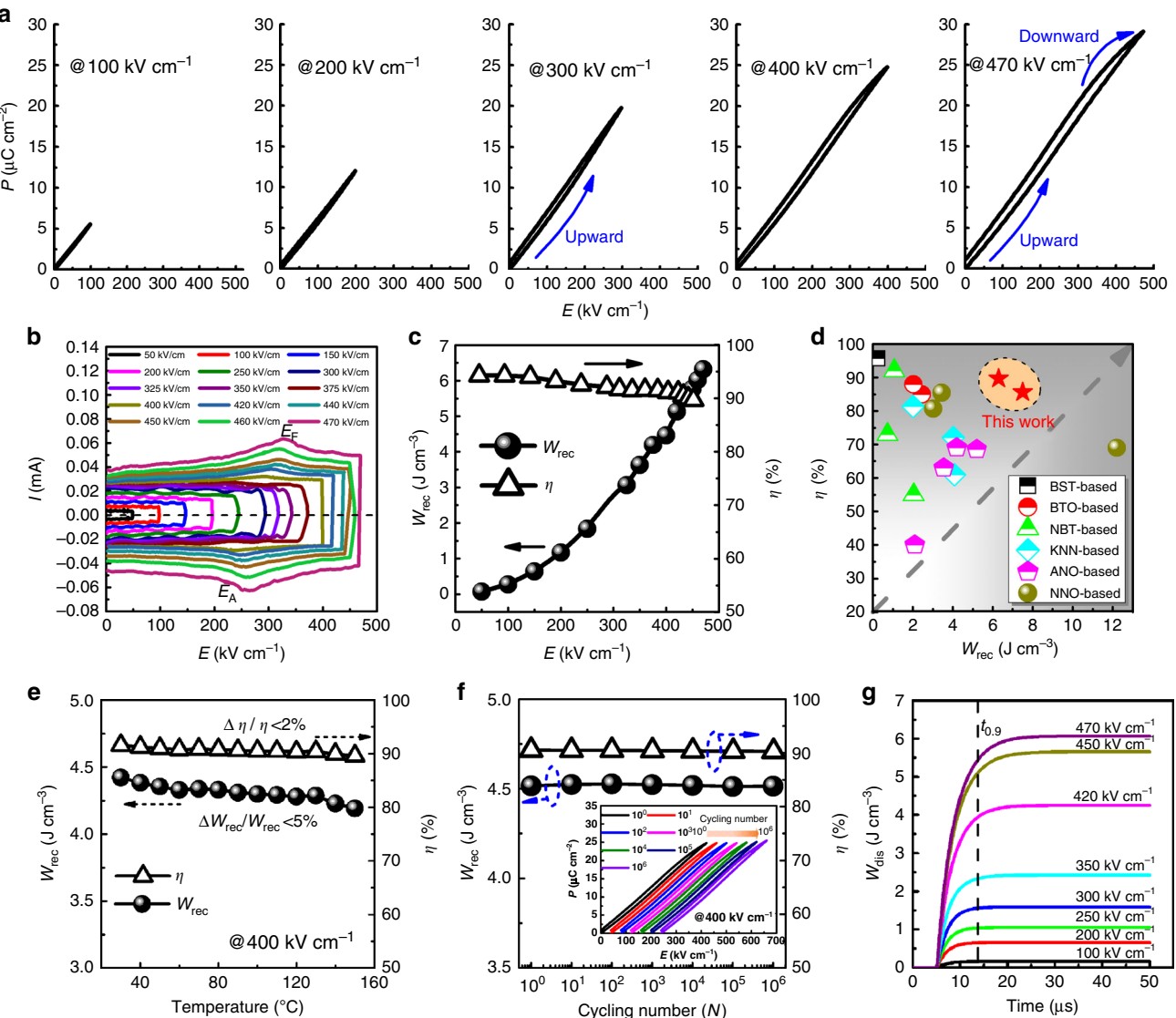

**Fig. 4 The reliability of energy storage performance under various conditions for the ANT55 ceramic with thickness of ~0.15 mm.** The electric field dependence of **a** $P–E$ loops, **b** $I–E$ curves, and **c** $W_{rec}$ and $\eta$ values. **d** A comparison of the energy storage properties of ANT55 and the state-of-the-art dielectric bulk ceramics. BST, BTO, NBT, KNN, ANO, and NNO represent $Sr_{0.7}Ba_{0.3}TiO_3$, $BaTiO_3$, $Bi_{0.5}Na_{0.5}TiO_3$, $K_{0.5}N_{0.5}NbO_3$, $AgNbO_3$, and $NaNbO_3$, respectively. **e** Temperature dependence of $W_{rec}$ and $\eta$ under an electric field of 400 kV cm$^{-1}$. **f** The room temperature $W_{rec}$ and $\eta$ as a function of the cycling number under an electric field of 400 kV cm$^{-1}$. The inset gives the typical $P–E$ loops after various cycling numbers. **g** The time dependence of discharge energy density under various electric fields, measured by an RC load circuit. The load resistance is 13 kΩ.

field, similar to those observed in antiferroelectric materials. With the increase of electric field, the $P–E$ loops gradually bend upward with remarkable enhancement of polarization, being associated with the AFE–FE phase transition, beyond which, the $P–E$ loops gradually bend downward, due to the saturated polarization in FE phase. The $P–E$ loops remain slim in shape with minimal hysteresis over the entire measuring electric field range, ascribing to the relaxor characteristics of ANT55. The AFE feature is further confirmed by the unipolar $I–V$ curves in Fig. 4b, in which two peaks are observed with the increase of electric field. These peaks are associated with the AFE-to-FE ($E_F$, ~330 kV cm$^{-1}$) and FE-to-AFE ($E_A$, ~270 kV cm$^{-1}$) phase transitions, respectively. The energy storage density also exhibits strong electric field dependent behavior, which increases significantly around phase transition (Fig. 4c), as generally observed in antiferroelectrics. The energy storage efficiency maintains high level of >90% over the measuring electric field. It should be noted that the breakdown strength is sample dimension dependent, where a higher

breakdown strength of 530 kV cm$^{-1}$ with the maximum polarization of 32 μC cm$^{-2}$ is achieved in sample with thickness of ~80 μm and electrode diameter of 2 mm. This leads to improved energy storage density up to 7.5 J cm$^{-3}$, meanwhile with yet high energy storage efficiency of 86% (Supplementary Fig. 4). Compared with state-of-the-art lead-free bulk ceramics[17,20,24,25,29,40–49], the ANT55 exhibits more attractive energy storage performance with both high energy storage density and efficiency, as shown in Fig. 4d.

From application viewpoint, the temperature stability and cycling reliability of energy storage properties are important[16,50]. The temperature-dependent energy storage property of ANT55 is evaluated at 400 kV cm$^{-1}$ to guarantee the safety in practical application, the corresponding results are given in Fig. 4e. The ANT55 exhibits very good temperature stability over temperature range of 25–150 °C, with minimal variations of <5 and <2% for energy storage density and efficiency respectively. Figure 4f gives the cycling reliability of ANT55. Both energy storage density and

efficiency maintain the same values at 400 kV cm$^{-1}$ after $10^6$ cycles, indicating the ANT55 has outstanding cycling reliability. In general, the large volume strain accompanied with AFE–FE phase transition in classic antiferroelectrics might accelerate the mechanical failure due to the electromechanical breakdown[51], while the strong relaxor feature in ANT55 smears the phase transition, leading to the highly improved cycling reliability[16].

In addition to the stability and reliability, the charge–discharge of dielectric capacitor is also important for high-power energy storage application. The charge–discharge performance of ANT55 is measured at RT using a resistance–capacitance (RC) circuit. The discharge energy density ($W_{dis}$) is calculated according to $W_{dis} = R \int i(t)^2 dt / V$[52], where $V$ is the sample volume and $R$ is the load resistor (13 k$\Omega$). The time dependence of $W_{dis}$ under various electric fields is displayed in Fig. 4g. The $W_{dis}$ is measured to be 6.1 J cm$^{-3}$ at 470 kV cm$^{-1}$, comparable to that calculated from $P$-$E$ loops. The small variation in values between two methods may be associated with the loss of discharged energy in the equivalent series resistor (ESR), domain walls movement and measurement frequency[16,53]. Moreover, the discharge time ($t_{0.9}$, 90% of all stored energy is released) is less than 15 µs, revealing a high discharge speed.

**Antiferroelectric ordering and local structure heterogeneity analysis.** The excellent energy storage properties in ANT$x$ solid solution are believed to be associated with its microstructure, i.e., the existence of AFE phase and relaxor component. In order to understand the relationship between the microstructure and energy storage properties, the synchrotron X-ray diffraction (SXRD), Raman spectra and annular dark-field scanning transmission electron microscopy (ADF-STEM) imaging were performed on the ANT$x$ samples. Figure 5a shows the SXRD of ANT$x$ ceramic powders, where pure perovskite structure can be observed for all ANT$x$ solid solutions. The (220) and (008) reflections appear as a single peak at $x > 50$ mol%. The corresponding Rietveld refinements of SXRD profiles based on $Pbcm$ space group are given in Supplementary Fig. 5. The low reliability factor values indicate the structural model is valid and the refinement results fit well with the experimental data. The Rietveld parameters are given in Fig. 5b–d, all of which exhibit strong composition-dependent behavior. The reduced cell volume might be attributed to the lower effective electronegativity of Ta$^{5+}$ (1.5) compared to that of Nb$^{5+}$ (1.6) since their ionic radii and valence are the same[54,55]. The displacements for Ag1 (located in 4d site in the $Pbcm$ structure, see Supplementary Fig. 6a) and B-site (Nb/Ta) cations are calculated and found to decrease with increasing Ta content (Fig. 5c), indicating a weaker ordering of local displacements[55]. The [Nb/TaO$_6$] octahedral tilting angles $\theta$ and $\Phi$ (the $\theta$ and $\Phi$ are tilting angles along b and c axes, respectively) also decrease with increase of Ta content (Fig. 5d), due to the smaller size of [TaO$_6$] octahedra. The decreased cell volumes, cation displacements and octahedral tilting angles are thought to reflect an improved stability of AFE phase[32,55]. Of particular significance is that the parameters exhibit different trends as function of Ta level over the range of 50 mol to 60 mol% (guided by the blue shaded area in Fig. 5). Similar phenomenon is also observed for the lattice parameters, (220)/(008) d-spacing and |B-O| distance (Supplementary Fig. 6b–d). The abnormal variations in parameters might be attributed to M2–M3 phase transition, as generally observed in pure AN counterpart around M2–M3 phase transition temperature[32], demonstrating that the $x = 50$–60 mol% compositions possess room temperature M2–M3 phase boundary. Figure 5e gives the Raman spectra of ANT$x$ solid solution, where the incorporation of Ta is found to shift Raman wavenumbers to lower values, due to the higher atomic mass of Ta comparing to that of Nb. All Raman peaks become weaker in intensity and broader in shape with the increase of Ta content,

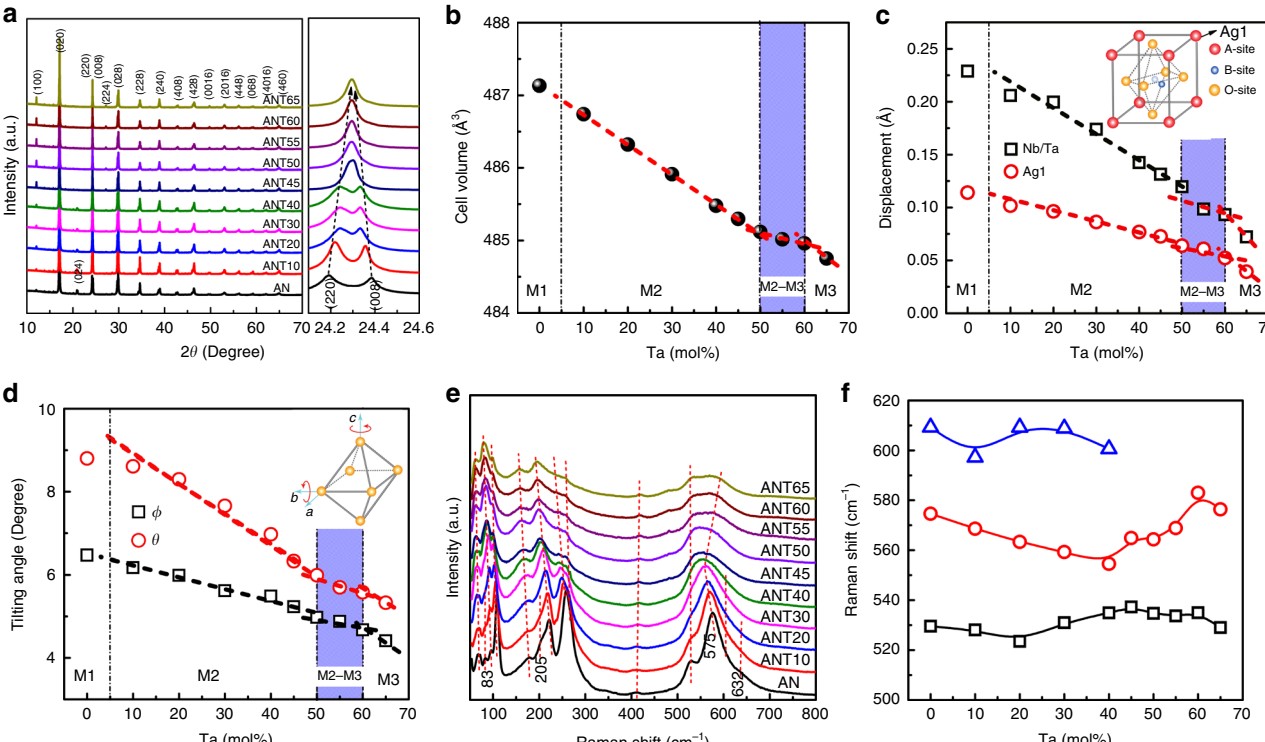

**Fig. 5 SXRD and Raman spectra of ANT$x$ ceramics. a** SXRD of the as-prepared ANT$x$ ceramic powders. The composition dependence of **b** cell volume, **c** displacement of Ag1 and Nb/Ta cations, and **d** the [Nb/TaO$_6$] octahedral tilting angles $\theta$ and $\Phi$ after refinement. **e** Raman spectra of the as-prepared ANT$x$ ceramics, and **f** the fitted Raman shifts for the peaks with wavenumber at 500–650 cm$^{-2}$.

being related to the improved disorder or relaxor feature. The peaks around 83, 205, and 632 cm$^{-1}$, which are associated with the M1–M2 (FIE–AFE) phase transition, disappear at $x = 20$ mol %, being consistent with the dielectric measurement. The peaks with wavenumber at 500–650 cm$^{-1}$ were fitted by using Gaussian function. Of particular interest is that the three fitted peaks in AN are merged into two peaks with Ta content over 40 mol% (Fig. 5f), revealing a possible M2–M3 phase transition above this composition, in agreement with the above dielectric and SXRD analysis regardless of the small deviation in composition.

The simultaneous integrated differential phase contrast (iDPC) and annular dark-field scanning transmission electron microscopy (ADF-STEM) imaging are performed on samples with $x = 0$ mol and 55 mol%, in order to investigate the local heterogeneity after Ta incorporation. While ADF imaging affords mass contrast imaging, the phase contrast of iDPC imaging is sensitive to light elements, allowing for the observation of oxygen[56]. Supplementary Fig. 7 shows atomic resolution iDPC images for AN and ANT55 samples on grains oriented along the [100]$_{pc}$ and [011]$_{pc}$ zone axes. Structural models generated from SXRD measurements are overlaid on the respective images illustrating the effect of octahedral distortion on the shape of the oxygen columns. Observations made directly from iDPC images are in agreement with SXRD measurements showing decreased octahedral distortion in ANT55 in comparison with AN. Figure 6a, b illustrates the oxygen–oxygen distances measured directly from iDPC images acquired for AN and ANT55 samples on grains oriented along the pseudocubic [011]$_{pc}$ zone axis, respectively. Obviously, the overall distortion on the oxygen sublattice decreases with mean oxygen–oxygen distances, which is 394.2 pm for AN and 393.1 pm for ANT55. It is important to note that while these mean values do not change significantly, the standard deviation of the distances decrease from 40 pm for AN to 25 pm for ANT55, in agreement with the expected decrease in octahedral tilting. The difference in cation–cation distances can also be determined for each sample. Cation–cation distances for the A sublattice (Ag) along the [110]$_{pc}$ direction are plotted in Fig. 6c, d for AN and ANT55 samples, respectively. Similar to O–O distances, the A–A distances are determined to decrease with Ta content, falling from 276.8 pm for AN to 276.2 pm for ANT55 samples. Conversely, while the A–A distances tend to decrease with Ta incorporation, the standard deviation of these distances increases from 2.9 to 4.5 pm between AN and ANT55, suggesting the increased local structural heterogeneity in ANT55 samples, induced by the mixing of Nb/Ta on the B sublattices, which is consistent with observations made in other relaxor systems[57]. To further demonstrate local structural heterogeneity, B sublattice (with respect to the A sublattice) displacements are plotted for the [100]$_{pc}$ zone axis as shown in Supplementary Fig. 8. For AN sample, a regular cation displacement pattern consistent with long-range antiferroelectricity is evident. For comparison, cation displacements for ANT55 vary significantly and lack long-range cooperation, being consistent with the scenario in a relaxor[58]. These observations provide strong evidence that Ta incorporation increases local structure heterogeneity in cation structure of ANT55. The existence of local structure heterogeneity in a fully stabilized AFE leads to relaxor AFEs and thus effectively impede the formation of macroscopic domain and smear the AFE–FE phase transition process. This is responsible for the nearly hysteresis-free P–E loops and the excellent energy storage properties in the designed ANT55 ceramics.

In summary, high energy storage density (6.3 J cm$^{-3}$) and efficiency (90%) are achieved simultaneously in 0.45AgNbO$_3$–0.55AgTaO$_3$ bulk ceramics, by judiciously constructing the diffused M2–M3 phase boundary. The material exhibits broad usage temperature range up to 150 °C, with minimal variations less than 5 and 2% for energy storage density and efficiency, respectively. Meanwhile the minimal variations in storage density and efficiency as function of cycling number up to 10$^6$ reveal excellent cycling reliability. All the merits demonstrate

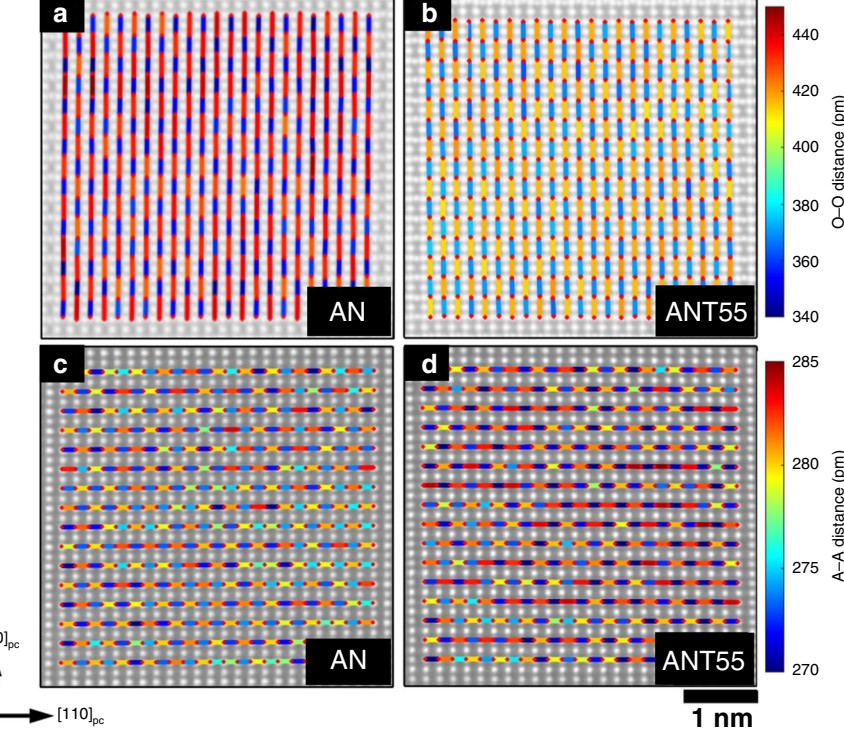

**Fig. 6 The oxygen–oxygen and A sublattice cation–cation distances of AN and ANT55 ceramics.** Oxygen–oxygen distances in the [100]$_{pc}$ direction for the **a** AN and **b** ANT55 samples. A sublattice cation–cation distances in the [110]$_{pc}$ direction for **c** AN and **d** ANT55 samples.

that $0.45AgNbO_3$–$0.55AgTaO_3$ ceramic is a promising candidate for high-power energy storage applications. It should be noticed that the energy storage density would be further improved in ANT$x$ multilayer ceramics and film capacitors, due to the significantly increased breakdown strength. In addition, the unfolding of RAFE characteristic of M2–M3 phase boundary on atomic scale in AN-based solid solution gives a solid evidence to the long-term confusion of the broad dielectric anomaly over M2–M3 phase transition temperature, which opens a broad range of applications where relaxor feature is desired, such as electrocaloric solid-state cooling devices[59,60] and hysteresis-free actuators[61,62].

## Methods

**Ceramic fabrication**. The $(1-x)AgNbO_3$–$xAgTaO_3$ ($x$ is 0, 10, 20, 30, 40, 45, 50, 55, 60, and 65 mol%, abbreviated as ANT$x$: ANT0-65) ceramics were synthesized by a conventional solid-state reaction method. The raw materials $Ag_2O$ (99.7%), $Nb_2O_5$ (99.5%), and $Ta_2O_5$ (99.99%) were carefully weighed and ball milled for 24 h in a nylon jar with alcohol using yttrium stabilized zirconia balls as milling media. The mixed powders were dried and then calcined at 900 °C for 6 h in oxygen atmosphere, followed by a second ball milling. The granulated powers were pressed into pellets with a diameter of 8 mm and thickness of ~1 mm, followed by cold isostatic pressing under 200 MPa to improve the green density. The pellets were then sintered at 1070–1180 °C based on the compositions for 6 h in oxygen atmosphere. For electrical properties measurement, the ceramics were polished down to a thickness of ~0.15 mm and then two parallel surfaces were coated with silver paste (~3 mm in diameter), and finally fired at 560 °C for 30 min as electrodes.

**Dielectric measurements**. The temperature dependence of dielectric permittivity and loss was measured using an LCR analyzer (Model 4294 A, Hewlett-Packard Co., Palo Alto, CA, USA) over the temperature range from −180 to 440 °C. The electric field dependence of normalized dielectric permittivity (dielectric tunability) was measured using an TF Analyzer 2000 (aixACCT, Aachen, Germany) with a maximum bias field of 80 kV cm$^{-1}$.

**Ferroelectric measurements**. $P$–$E$ loops and $I$–$V$ curves were measured under a triangular field at 1 Hz by using the ferroelectric testing system (Precision Multiferroic, Radiant Technologies Inc., Albuquerque, NM) connected to a homemade heating system.

**Charge–discharge measurements**. The discharge speed and discharge energy density were measured using a capacitor charge–discharge test system (PK-CPR1701, PolyK Technologies, PA, USA).

**Dielectric breakdown test**. The $E_b$ was measured using a voltage breakdown tester (RK2671AM, Shenzhen Meiruike electronic technology Co. Ltd, Shenzhen, China) on the sample with thickness of ~0.15 mm and diameter of 3 mm. The value of $E_b$ was evaluated by using the following Weibull distribution functions:[21,49]

$$X_i = \ln(E_i) \tag{1}$$

$$Y_i = \ln\{\ln[1/(1 - N_i)]\} \tag{2}$$

$$N_i = i/(n + 1) \tag{3}$$

where $n$ is the total number of samples, $E_i$ is the breakdown electric field for the $i$th specimen arranging in ascending order, and $N_i$ the probability of dielectric breakdown. $X_i$ and $Y_i$ should have a linear relationship.

**Band gap tests**. Ultraviolet and visible (UV–vis) absorption spectra was obtained using a UV–vis spectrometer (UV3600, Shimadzu, Kyoto, Japan), fitted with $BaSO_4$ as the standard material in the wavelength region of 200–800 nm.

**Characterization of phase and microstructure**. The microstructure of the polished and thermally etched samples was observed using a scanning electron microscope (Phenom Pro X, Phenom-World, Eindhoven, Netherlands). The crystal structure was characterized using synchrotron X-ray diffractions (SXRD) in a capillary mode. The high-resolution data were collected at TPS 09 A (Taiwan Photon Source) of the National Synchrotron Radiation Research Center. The 15 keV X-ray source (wavelength 0.826569 Å) is delivered from an in-vacuum undulator (IU22), and the powder diffraction patterns were recorded by a position-sensitive detector, MYTHEN 24 K, covering a $2\theta$ range of 120°. The full XRD data were analyzed by the Rietveld refinement using TOPAS 4.2 software (Bruker AXS GmbH, Germany). The Raman spectra was carried out using a laser confocal Raman microspectroscopy (LabRAM HR800, Horiba JobinYvo) with excitation at 532 nm and 50 mW. Atomic resolution scanning transmission electron microscope (STEM) was performed on an image and

probe corrected FEI Themis Z 60–300k kV S/TEM (ThermoFisher Scientific, Eindhoven, Netherlands) equipped with an X-FEG source and operated at an accelerating voltage of 300 kV. A beam current of 15 pA and a semi-angle of convergence of 17.9 mrad was utilized. ADF images were collected with a detector semi-angle range of 28–180 mrad while iDPC images were collected with a detector semi-angle range of 7–28 mrad. Distortion corrected images were produced via post processing two (1024 × 1024 pixel, 10 μs/pixel dwell time) images acquired at orthogonal scan directions[63]. Atom column locations were determined via Atomap[64] with analysis being performed with custom MATLAB and Python scripts.

## Data availability
The data that support the plots within this paper and other findings of this study are either provided in the Article and its Supplementary information or available from the corresponding author upon request.

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

## Acknowledgements

This work was financially supported by the National Natural Science Foundation of China (Grant No. 11864004), the Natural Science Foundation of Guangxi Province (Grant No. 2017GXNSFBA198132), the Science and Technology Major Project of Guangxi Province (Grant No. AA17204100). S.Z. acknowledges the support of the Australian Research Council (Grant No. FT140100698). M.J.C. and X.L. acknowledge the support of the Australian Research Council (Grant No. DP190101155). J.L. acknowledges the support of the Basic Science Center Program of the National Natural Science Foundation of China (Grant No. 51788104). We are grateful for the scientific and technical support from the Australian Centre for Microscopy and Microanalysis as well as the Microscopy Australia node the University of Sydney. We also acknowledge the support team (Dr. H.S. Sheu, Dr. Y.C. Chuang, Dr. Y.C. Lai, and Mr. C.K. Chang) for assistance at TPS 09 A in NSRRC.

## Author contributions

N.L. conceived the initial concept. N.L., H.K., X.C., and Q.F. prepared the sample and processed the experimental data. N.L., M.J.C., S.Z., Y.W., and J. L. interpreted the theoretical and experimental results. M.J.C. and X.L. conducted the TEM observations. C.L. measured the SXRD. G.Z. conducted the charge–discharge measurement. N.L., M.J.C., and S.Z. wrote the paper, all authors discussed and edited the paper.

## Competing interests

The authors declare no competing interests.
