## [Peer Review File · Nature Communications]

REVIEWER COMMENTS

Reviewer #1 (Remarks to the Author):

N. Luo and co-authors present high performance lead-free dielectric materials based on ANO AFE-ATaO solid solution system for high energy storage capacitor application. The authors introduce new material design concept i.e., highly stabilized AFE with relaxor feature to achieve both high energy storage density and high efficiency simultaneously. The obtained data are remarkable among lead-free based dielectric materials, and the proposed material design strategy, analysis and interpretation of the obtained data, material characterizations (phase, microstructure, etc.) are sound and not misleading. Though there are some reports on the ANO based AFE materials with dopant, the work presented in this study has a unique concept of AFE-RFE phase coexistence for enhanced energy storage and efficiency. Therefore, I recommend the publication of this manuscript. However bellow concerns should be considered before publication.

Page 6, after line 136, Authors mentioned that AFE-FE phase transition is shifted to higher E with the addition of Ta and this P-E loop evolution is the reveal of the improved stability of AFE phase. In addition, the authors argued in the Fig. SI 1, AFE is enhanced with increase of Ta content. But it seems like AFE is not stabilized, but more smeared as RFE characteristic. It should be more clearly addressed in the revised manuscript. I am not sure whether we can claim this as 'highly stabilized'.

Line 249 and 266, what is the meaning of Ag'1'?

Reviewer #2 (Remarks to the Author):

Authors have thoroughly reported High energy storage and efficiency in AgNbO₃-AgTaO₃ solid solution. I could find results interesting. However, I feel Nature Communication can not be appropriate platform for this manuscript as neither this topic/material is novel nor approach is unique as can be seen from reference list where AgNbO₃-AgTaO₃ is well explored.

Reviewer #3 (Remarks to the Author):

The present manuscript reports that high energy storage density (6.3 J cm⁻³) and efficiency (90%) can be simultaneously achieved by constructing a room temperature M₂-M₃ phase boundary in the Ag(NbTa)O₃ system. Moreover, the material was found to show high stability of energy storage density and efficiency over a wide temperature range and excellent cycling reliability. The excellent energy storage performance and stability are attributed to relaxor antiferroelectric features relevant to the local structure heterogeneity and antiferroelectric ordering, which are supported by scanning transmission electron microscopy and synchrotron x-ray diffraction. It is a solid work with systematic characterizations, measurements and reasonable discussions. However, I am not well convinced that the paper deserves publication in Nat. Communi. While the basic idea of using relaxor antiferroelectrics (highly stabilized antiferroelectricity with relaxor feature) to develop dielectric capacitors with high energy storage performance has already been demonstrated in several solid-solution systems, e.g., (Bi_{0.5}Na_{0.5})TiO₃-NaNbO₃ (Adv. Funct. Mater. 2019, 29, 1903877), and

(Na_{0.5}Bi_{0.5})TiO₃-x(Sr_{0.7}Bi_{0.2})TiO₃ (Adv. Mater. 2018, 30, 1802155), the superior energy storage performance of the Ag(NbTa)O₃ system has also been explored and reported (e.g., Adv. Mater. 2017, 1701824). The key point/novelty of the present work could be determining of the room temperature M₂-M₃ phase boundary, i.e. Ag(Nb_{0.45}Ta_{0.55})O₃, but this info was already well documented in the literature (J. Eur. Ceram. Soc. 27 (2007) 2549-2560; from Fig. 1, the room temperature M₂-M₃ phase boundary is known around Ta content of 0.55). The present work can be regarded as an extension of previous work (Adv. Mater. 2017, 1701824), and verifying that Ag(Nb_{0.45}Ta_{0.55})O₃ is in the M₂-M₃ phase boundary at room temperature, and possesses enhanced relaxor antiferroelectric behavior, and thus has further improved energy storage performance. In addition, I have the following points that need author's attention.

1. P4/lines 92-93: "the M₂-M₃ phase boundary is shifted downward to room temperature"; P4/lines 104-105: "the M₂-M₃ phase boundary was shifted to room temperature": in these sentences, "M₂-M₃ phase boundary" is mixed up with "M₂-M₃ phase transition". While the M₂-M₃ phase boundary is around 0.55-0.6, the M₂-M₃ phase transition takes place at room temperature (RT); in other word, when the Ta content is around 0.55-0.6 in Ag(NbTa)O₃, the M₂-M₃ phase transition temperature is RT (i.e., the M₂ and M₃ phases are coexisting at RT).
2. Why do the peaks around 575 cm⁻¹ firstly shift to lower wavenumber and then go upward at around 45 mol%? The author attributed it to a possible M₂-M₃ phase transition that happened in the ANTx ceramics. The article would be more interesting if the detailed M₂-M₃ phase transition process was analyzed. As M₂-M₃ phase transition at RT (or M₂ and M₃ phases coexisting at RT) is the key for the enhanced relaxor antiferroelectric behavior of Ag(Nb_{0.45}Ta_{0.55})O₃, this is of most interest to see whether it is indeed the order-disorder or alternatively composition inhomogeneity responsible for the relaxor behavior.
3. As known, dielectric thin films and nanocomposites have been widely investigated in the field of energy storage recently. Can the authors comment on the advantages of ANTx solid solution for application in energy storage compared with the film and nanocomposite materials?
4. The two arrow indicators in Fig. 3b are probably mixed up.

Reviewer #4 (Remarks to the Author):

Authors presented an excellent work entitled "Constructing phase boundary in AgNbO₃ antiferroelectrics: pathway simultaneously achieving high energy density and efficiency"

This work provides much awaited breakthrough in high energy density and high efficiency capacitor material for energy storage. This is specially important for futuristic miniaturized devices. The developed material clearly exhibit not only high temperature stability but also high cycle stability. Furthermore, this work provide fundamental understanding of the phenomenon, which could be used as guiding principle for developing such materials for various applications. This paper is suitable for nature communications.

Here is some suggestions for further improving manuscript:

1. It will be great if authors could generate a graph showing their results versus other reports on the same topic.
2. Please add some recent and more relevant references. For example:
(a) Relaxor behavior and electrothermal properties of Sn- and Nb-modified (Ba,Ca)TiO₃ Pb-free

ferroelectric.

(b) A new method for achieving enhanced dielectric response over a wide temperature range

Reviewers' comments:

Reviewer #1 (Remarks to the Author):

N. Luo and co-authors present high performance lead-free dielectric materials based on ANO AFE-ATaO solid solution system for high energy storage capacitor application. The authors introduce new material design concept i.e., highly stabilized AFE with relaxor feature to achieve both high energy storage density and high efficiency simultaneously. The obtained data are remarkable among lead-free based dielectric materials, and the proposed material design strategy, analysis and interpretation of the obtained data, material characterizations (phase, microstructure, etc.) are sound and not misleading. Though there are some reports on the ANO based AFE materials with dopant, the work presented in this study has a unique concept of AFE-RFE phase coexistence for enhanced energy storage and efficiency. Therefore, I recommend the publication of this manuscript. However bellow concerns should be considered before publication.

1. Page 6, after line 136, Authors mentioned that AFE-FE phase transition is shifted to higher E with the addition of Ta and this P-E loop evolution is the reveal of the improved stability of AFE phase. In addition, the authors argued in the Fig. SI 1, AFE is enhanced with increase of Ta content. But it seems like AFE is not stabilized, but more smeared as RFE characteristic. It should be more clearly addressed in the revised manuscript. I am not sure whether we can claim this as 'highly stabilized'.

2. Line 249 and 266, what is the meaning of Ag¹'?

Reviewer #2 (Remarks to the Author):

Authors have thoroughly reported High energy storage and efficiency in AgNbO₃-AgTaO₃ solid solution. I could find results interesting. However, I feel Nature Communication can not be appropriate platform for this manuscript as neither this topic/material is novel nor approach is unique as can be seen from reference list where AgNbO₃-AgTaO₃ is well explored.

Reviewer #3 (Remarks to the Author):

The present manuscript reports that high energy storage density (6.3 J cm^{-3}) and efficiency (90%) can be simultaneously achieved by constructing a room temperature M2–M3 phase boundary in the $\text{Ag}(\text{NbTa})\text{O}_3$ system. Moreover, the material was found to show high stability of energy storage density and efficiency over a wide temperature range and excellent cycling reliability. The excellent energy storage performance and stability are attributed to relaxor antiferroelectric features relevant to the local structure heterogeneity and antiferroelectric ordering, which are supported by scanning transmission electron microscopy and synchrotron x-ray diffraction. It is a solid work with systematic characterizations, measurements and reasonable discussions. However, I am not well convinced that the paper deserves publication in Nat. Communi. While the basic idea of using relaxor antiferroelectrics (highly stabilized antiferroelectricity with relaxor feature) to develop dielectric capacitors with high energy storage performance has already been demonstrated in several solid-solution systems, e.g., $(\text{Bi}_{0.5}\text{Na}_{0.5})\text{TiO}_3\text{-NaNbO}_3$ (Adv. Funct. Mater. 2019, 29, 1903877), and $(\text{Na}_{0.5}\text{Bi}_{0.5})\text{TiO}_3\text{-x}(\text{Sr}_{0.7}\text{Bi}_{0.2})\text{TiO}_3$ (Adv. Mater. 2018, 30, 1802155), the superior energy storage performance of the $\text{Ag}(\text{NbTa})\text{O}_3$ system has also been explored and reported (e.g., Adv. Mater. 2017, 1701824). The key point/novelty of the present work could be determining of the room temperature M2–M3 phase boundary, i.e. $\text{Ag}(\text{Nb}_{0.45}\text{Ta}_{0.55})\text{O}_3$, but this info was already well documented in the literature (J. Eur. Ceram. Soc. 27 (2007) 2549–2560; from Fig. 1, the room temperature M2–M3 phase boundary is known around Ta content of 0.55) . The present work can be regarded as an extension of previous work (Adv. Mater. 2017, 1701824), and verifying that $\text{Ag}(\text{Nb}_{0.45}\text{Ta}_{0.55})\text{O}_3$ is in the M2–M3 phase boundary at room temperature, and possesses enhanced relaxor antiferroelectric behavior, and thus has further improved energy storage performance. In addition, I have the following points that need author's attention.

1. P4/lines 92-93: “the M2–M3 phase boundary is shifted downward to room temperature”; P4/lines 104-105: “the M2–M3 phase boundary was shifted to room temperature”: in these sentences, “M2–M3 phase boundary” is mixed up with “M2–

M3 phase transition". While the M2–M3 phase boundary is around 0.55-0.6, the M2–M3 phase transition takes place at room temperature (RT); in other word, when the Ta content is around 0.55-0.6 in Ag(NbTa)O₃, the M2–M3 phase transition temperature is RT (i.e., the M2 and M3 phases are coexisting at RT).

2. Why do the peaks around 575 cm⁻¹ firstly shift to lower wavenumber and then go upward at around 45 mol%? The author attributed it to a possible M2–M3 phase transition that happened in the ANT_x ceramics. The article would be more interesting if the detailed M2–M3 phase transition process was analyzed. As M2-M3 phase transition at RT (or M2 and M3 phases coexisting at RT) is the key for the enhanced relaxor antiferroelectric behavior of Ag(Nb_{0.45}Ta_{0.55})O₃, this is of most interest to see whether it is indeed the order-disorder or alternatively composition inhomogeneity responsible for the relaxor behavior.

3. As known, dielectric thin films and nanocomposites have been widely investigated in the field of energy storage recently. Can the authors comment on the advantages of ANT_x solid solution for application in energy storage compared with the film and nanocomposite materials?

4. The two arrow indicators in Fig. 3b are probably mixed up.

Reviewer #4 (Remarks to the Author):

Authors presented an excellent work entitled "Constructing phase boundary in AgNbO₃ antiferroelectrics: pathway simultaneously achieving high energy density and efficiency"

This work provides much awaited breakthrough in high energy density and high efficiency capacitor material for energy storage. This is specially important for futuristic miniaturized devices. The developed material clearly exhibit not only high temperature stability but also high cycle stability. Furthermore, this work provide fundamental understanding of the phenomenon, which could be used as guiding principle for developing such materials for various applications. This paper is suitable for nature communications.

Here is some suggestions for further improving manuscript:

1. It will be great if authors could generate a graph showing their results versus other reports on the same topic.
2. Please add some recent and more relevant references. For example:
 - (a) Relaxor behavior and electrothermal properties of Sn- and Nb-modified (Ba,Ca)TiO₃ Pb-free ferroelectric.
 - (b) A new method for achieving enhanced dielectric response over a wide temperature range

We thank the reviewers for their valuable reviews, insightful comments and constructive suggestions. We appreciate their positive comments, such as “obtained data are remarkable,” “solid work with systematic characterizations, measurements and reasonable discussions,” “specially important for futuristic miniaturized devices,” and “breakthrough.” We have revised the manuscript and Supplementary Information accordingly.

Below, we present point-by-point responses to the reviewers’ comments. Our responses to the reviewers’ comments are highlighted in blue. In the revised manuscript, we highlighted our modifications in red.

Point-by-point responses to the reviewers' comments

Reviewer #1 (Remarks to the Author):

N. Luo and co-authors present high performance lead-free dielectric materials based on ANO-ATaO solid solution system for high energy storage capacitor application. The authors introduce new material design concept i.e., highly stabilized AFE with relaxor feature to achieve both high energy storage density and high efficiency simultaneously. The obtained data are remarkable among lead-free based dielectric materials, and the proposed material design strategy, analysis and interpretation of the obtained data, material characterizations (phase, microstructure, etc.) are sound and not misleading. Though there are some reports on the ANO based AFE materials with dopant, the work presented in this study has a unique concept of AFE-RFE phase coexistence for enhanced energy storage and efficiency. Therefore, I recommend the publication of this manuscript. However bellow concerns should be considered before publication.

Comment 1: Page 6, after line 136, Authors mentioned that AFE-FE phase transition is shifted to higher E with the addition of Ta and this P-E loop evolution is the reveal of the improved stability of AFE phase. In addition, the authors argued in the Fig. SI 1, AFE is enhanced with increase of Ta content. But it seems like AFE is not stabilized, but more smeared as RFE characteristic. It should be more clearly addressed in the revised manuscript. I am not sure whether we can claim this as 'highly stabilized'.

Response:

We really appreciate the reviewer's positive comment. In the manuscript, we used the phrase "improved stability of AFE phase" based on the reasons listed below. Firstly, the M1-M2 phase transition temperature shifts to below room temperature, while the M2-M3 phase transition temperature also shifts downward to room temperature with increase of Ta content, which indicates that the AFE (M2 phase) region is expanded to room temperature. Secondly, the AFE-FE phase transition electric field in the *P-E* loops is increased obviously with increase of Ta content, which is generally observed in an AFE material with increased stability of AFE phase (need higher energy transform AFE to FE phase). Thirdly, electric field dependence of dielectric permittivity variation becomes more flattened with the increase of Ta

concentration (see Supplementary Fig. 1), which has been demonstrated in other AFE materials. From the microstructure perspective, the decreased ionic displacements with the increase of Ta content, obtained from the structure refinement, also give the evidence of enhanced stability of AFE phase. All the phenomena above reveals the increased stability of AFE phase by increasing Ta content. It should be admitted that the relaxor characteristic cause the slanted and slim P - E loops, which also leads to the enhanced AFE-FE phase transition electric field in a certain degree, due to the disruption of the long-range ordered AFE domains and the weak interatomic interactions between the AFE nanodomain clusters. However, in our case, obvious improvement in AFE-FE phase transition electric field also occurs for a low Ta content, where no relaxor characteristic is observed. So, we prefer not ascribing the enhancement of AFE-FE phase transition electric field to the relaxor characteristic, notwithstanding relaxor characteristic also play a role in the enhanced AFE stability. Based on the above consideration and knowledge, we used the phrase “highly stabilized”. To make it more clearly to the readers, the reasons are also addressed in the revised manuscript.

Comment 2: Line 249 and 266, what is the meaning of Ag'1'?

Response: Thanks for the question and sorry about the confusion. At room temperature, AgNbO_3 and $\text{AgNbO}_3\text{-AgTaO}_3$ solid solution generally possess perovskite structure that exhibits orthorhombic symmetry with Pbcm space group. This superstructure features a tilting of $[\text{NbO}_6]$ octahedra described in Glazer's notation as $a^-b^-c^-/a^-b^-c^+$. A sequence of two in-phase and two antiphase octahedral rotating around the c -axis produces $4a_c$ periodicity, in which the two Ag sites (Ag1 in 4d site, Ag2 in 4c site) are different in crystal structure, as list in Table 1. Furthermore, the Ag1 and Ag2 ions shift along different directions, with Ag1 and Ag2 shifting along the b axis and a axis, respectively (Fig. 1).

Table 1 The structure parameters of Ag1 and Ag2 in AgNbO_3 with Pbcm space group.

Atom	Site	x	y	z	Displacement (\AA)
Ag1	4d	0.7545(6)	0.236(5)	0.75	0.135(3)
Ag2	4c	0.730(4)	0.25	0.5	0.019(6)

Fig. 1. Schematics of the Ag and Nb displacements in the Pbcm structure. The displacement directions are indicated using arrows. (This figure can be seen as Supplementary Fig. 6 (a) in the revised manuscript.)

Reviewer #2 (Remarks to the Author):

Comment: Authors have thoroughly reported high energy storage and efficiency in $\text{AgNbO}_3\text{-AgTaO}_3$ solid solution. I could find results interesting. However, I feel Nature Communication can not be appropriate platform for this manuscript as neither this topic/material is novel nor approach is unique as can be seen from reference list where $\text{AgNbO}_3\text{-AgTaO}_3$ is well explored.

Response:

We thank the reviewer for considering our finding interesting. However, we respectfully disagree with the referee's statement that "neither this topic/material is novel nor approach is unique as can be seen from reference list where $\text{AgNbO}_3\text{-AgTaO}_3$ is well explored". Here in the following we would like to clarify the significance and highlights of our research to justify the publication of our paper in Nature Communication.

Because of the great advantages of dielectric energy storage capacitors (high power density, ultra-fast charge and discharge speeds, long-term usage, etc.) compared to electrochemical energy storage systems (battery, super capacitor, etc.), dielectric capacitors have attracted great attention in the last decades. Actually, both the published papers and citations in the past 10 years have experienced substantial growth when use "dielectric energy storage" as the theme in Web of Science, as summarized in Fig.2. In 2019 alone, over 1000 papers were published with more than 30000 citations. All the statistics indicate the "dielectric energy storage" is a hot topic now and it is attracting extensive attention. Moreover, a large number of papers were published on high-impact journals, such as Science, Nature Materials, Nature Communications, Advanced Materials, Advanced Functional Materials, Energy & Environmental Science, etc.

We admitted that AgNbO_3 and $\text{AgNbO}_3\text{-AgTaO}_3$ solid solution might not be new materials. As early as 1958, Reisman and Holtzberg, Francombe and Lewis independently published the first papers, reporting the heterogeneous equilibria in $\text{Ag}_2\text{O-Nb}_2\text{O}_5$ system, the structure and dielectric properties of AgNbO_3 , AgTaO_3 and $\text{AgNbO}_3\text{-AgTaO}_3$ ceramics. (Reisman et al., J. Am. Chem. Soc., 1958, 80, 6503; Francombe et al., Acta Cryst., 1958, 11,175.) Later on, researchers studied the AgNbO_3 and the related materials from different perspectives, mainly focusing on crystal structure, dielectric properties, microwave dielectric applications, identifying

the ferroelectric/antiferroelectric characteristics, etc. In fact, all the published papers are less than 200 prior to 2016 (use “AgNbO₃” or “silver niobate” as the theme in Web of Science). A breakthrough was made in 2016, when the unique antiferroelectric double hysteresis loops and high-energy storage density were established in pure AgNbO₃ ceramics, induced by high electric field over 150kV/cm benefiting from the improved quality of bulk ceramics. (Tian et al., J. Mater. Chem. A, 2016, 4, 17279-17287; Zhao et al., J. Mater. Chem. C, 2016, 4, 8380-8384) After that, intensive investigations have been focusing on the energy storage properties of AgNbO₃ based materials (Zhao et al., Adv. Mater., 2017, 29, 1701824; Luo et al., J. Mater. Chem. A, 2019, 7, 14118.), photoelectric and photovoltaic effect (Liu et al., Adv. Funct. Mater. 2019, 29, 1900918), to explosive energy conversion (Liu et al., Sci. Adv. 2020, 6, eaba0367). The significantly increased number of high-impact papers demonstrate AgNbO₃ and the related materials are attracting extensive attentions for emerging applications.

The dielectric anomalies and structure evolution of M1, M2, and M3 phases have been intensively investigated in AgNbO₃ based materials with temperature. Nevertheless, researchers are still struggling for giving a clear explanation for the structure origin of the broad dielectric anomaly of the M2-M3 phase transition. This greatly limits the application of AgNbO₃ with M2-M3 phase boundary. In this work, the atomic scale displacements disorder was discovered around the M2-M3 phase boundary, which is considered to be the origin of relaxor behavior. *This is the first time to give an atomic scale explanation for the relaxor characteristic of M2-M3 phase boundary.* Furthermore, the atomic scale displacements associated with the structure refinement from SXRD verify the highly stabilized AFE phase around the M2-M3 phase boundary. *The comprehension of the atomic scale structure may open new applications for AgNbO₃-based materials with the M2-M3 phase boundary.*

In very recent years, AgNbO₃ based AFE materials have been studied to achieve high energy density and efficiency, the previous studies were mainly based on the concept of stabilizing AFE phase. This is well confirmed in the A/B-site doped AgNbO₃ (Zhao et al., Adv. Mater. 2017, 1701824; Luo et al., J. Mater. Chem. A, 2019, 7, 14118), but with the clear disadvantage of low energy efficiency (less than 70%). On the other hand, the concept of relaxor antiferroelectric has been demonstrated in some lead free systems, however either of them was achieved by shifting the

antiferroelectric-paraelectric phase transition temperature (T_m) to room temperature, or forming less-stabilized antiferroelectricity, which are still unable to simultaneously achieve both high energy density and efficiency.

Based on the above structure analysis, we developed the idea of shifting the M2-M3 phase boundary in $\text{AgNbO}_3\text{-AgTaO}_3$ solid solution to room temperature, by taking the advantages of highly stabilized antiferroelectric feature and relaxor behavior. As a result, ultrahigh energy storage density and efficiency are achieved in this work. The significance of this work not only lies in the high energy storage density and efficiency, but also unfolding the relaxor characteristic of M2-M3 phase boundary on atomic scale in AgNbO_3 based solution, opening a new direction tuning the composition for specific applications.

Fig. 2 (a) Publications and (b) citations on dielectric energy storage in the past 10 year.

Reviewer #3 (Remarks to the Author):

Comment 1: The present manuscript reports that high energy storage density (6.3 J cm^{-3}) and efficiency (90%) can be simultaneously achieved by constructing a room temperature M2–M3 phase boundary in the $\text{Ag}(\text{NbTa})\text{O}_3$ system. Moreover, the material was found to show high stability of energy storage density and efficiency over a wide temperature range and excellent cycling reliability. The excellent energy storage performance and stability are attributed to relaxor antiferroelectric features relevant to the local structure heterogeneity and antiferroelectric ordering, which are supported by scanning transmission electron microscopy and synchrotron x-ray diffraction. It is a solid work with systematic characterizations, measurements and reasonable discussions. However, I am not well convinced that the paper deserves publication in Nat. Communi. While the basic idea of using relaxor antiferroelectrics (highly stabilized antiferroelectricity with relaxor feature) to develop dielectric capacitors with high energy storage performance has already been demonstrated in several solid-solution systems, e.g., $(\text{Bi}_{0.5}\text{Na}_{0.5})\text{TiO}_3\text{-NaNbO}_3$ (Adv. Funct. Mater. 2019, 29, 1903877), and $(\text{Na}_{0.5}\text{Bi}_{0.5})\text{TiO}_3\text{-x}(\text{Sr}_{0.7}\text{Bi}_{0.2})\text{TiO}_3$ (Adv. Mater. 2018, 30, 1802155), the superior energy storage performance of the $\text{Ag}(\text{NbTa})\text{O}_3$ system has also been explored and reported (e.g., Adv. Mater. 2017, 1701824). The key point/novelty of the present work could be determining of the room temperature M2–M3 phase boundary, i.e. $\text{Ag}(\text{Nb}_{0.45}\text{Ta}_{0.55})\text{O}_3$, but this info was already well documented in the literature (J. Eur. Ceram. Soc. 27 (2007) 2549–2560; from Fig. 1, the room temperature M2–M3 phase boundary is known around Ta content of 0.55) . The present work can be regarded as an extension of previous work (Adv. Mater. 2017, 1701824), and verifying that $\text{Ag}(\text{Nb}_{0.45}\text{Ta}_{0.55})\text{O}_3$ is in the M2–M3 phase boundary at room temperature, and possesses enhanced relaxor antiferroelectric behavior, and thus has further improved energy storage performance. In addition, I have the following points that need author’s attention.

Response:

We thank the reviewer for the positive comment of “It is a solid work with systematic characterizations, measurements and reasonable discussions”, also appreciate for providing many valuable references.

Here we would like to clarify the significance of our research. *We simultaneously achieved high energy storage density (6.3 J cm^{-3}) and efficiency (90%) by*

constructing a room temperature M2–M3 phase boundary in the Ag(NbTa)O₃ system. The M2–M3 phase boundary exhibits highly stable antiferroelectricity associated with good relaxor behavior which have never been studied. In addition to the good properties achieved, we provide a good paradigm for designing high-performance material on atomic scale to tailor the properties for specific applications.

We admitted that the concept of using relaxor antiferroelectric material to improve the energy density and efficiency has been demonstrated in several solid-solution systems, such as (Bi_{0.5}Na_{0.5})TiO₃-NaNbO₃ (Qi et al., Adv. Funct. Mater. 2019, 29, 1903877), and (Na_{0.5}Bi_{0.5})TiO₃-(Sr_{0.7}Bi_{0.2})TiO₃ (Li et al., Adv. Mater. 2018, 30, 1802155). However, the approach to realize this concept is different from ours, where the previous studies mainly concentrated on introducing relaxor behavior in antiferroelectric material, regardless the stability of antiferroelectric phase. This usually results in the quick saturation of polarization (due to the low antiferroelectric-ferroelectric transition electric field) or large hysteresis, leading to limited energy density or low efficiency. Considering the drawbacks of previous studies, we proposed a novel idea of developing high performance relaxor antiferroelectrics with both highly stabilized antiferroelectric phase and relaxor component, by shifting the M2-M3 phase boundary to room temperature. In this work, the antiferroelectric M2 and M3 phases coexist in the newly designed relaxor antiferroelectric, which possess highly stabilized antiferroelectric phase with relaxor characteristic. *The above-mentioned characteristics lead to the high energy density and efficiency simultaneously.*

Another highlight of this work is the atomic scale exploration of the structure evolution of M2-M3 phase boundary, which successfully explain the relaxor antiferroelectric feature. In 1958, Francombe and Lewis first investigated the phase structure transitions and the associated dielectric anomalies of AgNbO₃ and AgNbO₃-AgTaO₃ solid solutions. (Francombe et al., Acta Cryst., 1958, 11,175.) They found a series of dielectric anomalies with increase of temperature, but could not distinguish the structure evolution from XRD refinement. Until 1983, Łukaszewski et al. observed a series of phase transitions in AgNbO₃ single crystal by combining XRD, DTA and dielectric measurement, in which they considered the now called M1, M2 and M3 phases to be monoclinic distortion. (Łukaszewski et al., Phase Transitions, 1983, 3, 247.) In 1987, Pawelczyk prepared a series of AgNbO₃-AgTaO₃ solid

solutions with various AgTaO_3 component and successfully shifted the M2-M3 phase transition temperature to room temperature. (Pawelczyk et al., Phase Transitions, 1987, 8, 273.) The room temperature M2-M3 phase boundary was also documented in the literature. (Valant et al., J. Eur. Ceram. Soc. 2007, 27, 2549). However, despite the intensive investigations, researchers are struggling to give a reasonable explanation for the origin of the broad dielectric anomaly over the M2-M3 phase transition, especially for the possible relaxor behavior. In this work, the atomic scale displacements disorder was discovered around the M2-M3 phase boundary, which is considered to be the origin of relaxor behavior. Furthermore, the atomic scale displacements associated with the structure refinement from SXRD verify the highly stabilized AFE phase around the M2-M3 phase boundary. *This is the first time to give an atomic scale explanation for the relaxor characteristic and verify the antiferroelectric feature of M2-M3 phase boundary, which has been a long-term challenge.* In addition to the energy storage application in this work, the understanding of the atomic scale structure of M2-M3 phase boundary may open more applications for AgNbO_3 -based materials.

Therefore, the significance and highlights of this work are not only on the high energy storage density and efficiency by employing the M2-M3 phase boundary, but also unfolding the relaxor antiferroelectric characteristic of M2-M3 phase boundary on atomic scale in AgNbO_3 based solid solution, which are important for both theoretical studies practical applications.

Comment 2: P4/lines 92-93: “the M2-M3 phase boundary is shifted downward to room temperature”; P4/lines 104-105: “the M2-M3 phase boundary was shifted to room temperature”: in these sentences, “M2-M3 phase boundary” is mixed up with “M2-M3 phase transition”. While the M2-M3 phase boundary is around 0.55-0.6, the M2-M3 phase transition takes place at room temperature (RT); in other word, when the Ta content is around 0.55-0.6 in $\text{Ag}(\text{NbTa})\text{O}_3$, the M2-M3 phase transition temperature is RT (i.e., the M2 and M3 phases are coexisting at RT).

Response:

Thanks for the constructive suggestion. We are sorry about the mistake and confusion. We actually mixed up “phase boundary” with “phase transition” in the previous manuscript. The sentence “the M2-M3 phase boundary was shifted to room

temperature” is changed to “the M2–M3 phase transition temperature was shifted to RT” in the revised manuscript.

Comment 3: Why do the peaks around 575 cm^{-1} firstly shift to lower wavenumber and then go upward at around 45 mol%? The author attributed it to a possible M2–M3 phase transition that happened in the ANTx ceramics. The article would be more interesting if the detailed M2–M3 phase transition process was analyzed. As M2-M3 phase transition at RT (or M2 and M3 phases coexisting at RT) is the key for the enhanced relaxor antiferroelectric behavior of $\text{Ag}(\text{Nb}_{0.45}\text{Ta}_{0.55})\text{O}_3$, this is of most interest to see whether it is indeed the order-disorder or alternatively composition inhomogeneity responsible for the relaxor behavior.

Response:

Thanks for the valuable and constructive suggestion. The structure evolution of M2-M3 phase transition is actually very important in this work, as it is closely related to the antiferroelectricity and relaxor characteristic that contribute to the achieved promising properties. To further understand this part, we added more careful and detailed analysis on the SXR and Raman spectra.

Fig. 3a shows the SXR of ANTx ceramic powders, where pure perovskite structure can be observed for all ANTx solid solutions. The (220) and (008) reflections appear as a single peak at $x > 50$ mol%. The SXR profiles are Rietveld refined based on *Pbcm* space group, which show good reliability factor values. The Rietveld parameters are given in Figs. 3 and 4, all of which exhibit strong composition dependent behavior. The reduced cell volume might be attributed to the lower effective electronegativity of Ta^{5+} (1.5) compared to that of Nb^{5+} (1.6) since their ionic radii and valence are the same. (Eng et al., J. Solid State Chem. 2003, 175, 94; Levin et al. Chem. Mater. 2010, 22, 4987.) The displacements for Ag1 (located in 4d site in the *Pbcm* structure, see Fig. 4a) and B-site (Nb/Ta) cations are calculated and found to decrease with increasing Ta content (Fig. 3c), indicating a weaker ordering of local displacements. (Levin et al. Chem. Mater. 2010, 22, 4987.) The $[\text{Nb}/\text{TaO}_6]$ octahedral tilting angles θ and Φ (the θ and Φ are tilting angles along b and c axes, respectively) also decrease with increase of Ta content (Fig. 3d), due to the smaller size of $[\text{TaO}_6]$ octahedra. The decreased cell volumes, cation displacements and octahedral tilting angles are thought to reflect an improved stability of AFE phase. Of

particular significance is that the parameters exhibit different trends as function of Ta level over the range of 50 mol% to 60 mol% (guided by the blue shaded area in Figs. 3 and 4). Similar phenomenon is also observed for the lattice parameters, (220)/(008) d-spacing and |B-O| distance (Figs. 4b-d). The abnormal variations in parameters might be attributed to M2–M3 phase transition, as generally observed in pure AN counterpart around M2–M3 phase transition temperature, (Levin et al., *Phy. Rev. B*, 2009, 79, 104113.) demonstrating that the $x=50-60\text{mol}\%$ compositions possess room temperature M2–M3 phase boundary. Fig. 3e gives the Raman spectra of ANTx solid solution, where the incorporation of Ta is found to shift Raman wavenumbers to lower values, due to the higher atomic mass of Ta comparing to that of Nb. All Raman peaks become weaker in intensity and broader in shape with the increase of Ta content, being related to the improved disorder or relaxor feature. The peaks around 83, 205 and 632 cm^{-1} , which are associated with the M1–M2 (FIE–AFE) phase transition, disappear at $x=20\text{mol}\%$, being consistent with the dielectric measurement. The peaks with wavenumber at 500–650 cm^{-1} were fitted by using Gaussian function. Of particular interest is that the three fitted peaks in AN are merged into two peaks with Ta content over 40 mol% (Fig. 3f), revealing a possible M2–M3 phase transition above this composition, in agreement with the dielectric and SXRD analysis regardless of the small deviation in composition.

The discussions can also be found in the revised manuscript.

Fig. 3 (a) SXRD of the as-prepared ANTx ceramic powders. The composition dependence of

(b) cell volume, (c) displacement of Ag1 and Nb/Ta cations, and (d) the $[\text{Nb}/\text{TaO}_6]$ octahedral tilting angles θ and Φ after refinement. (e) Raman spectra of the as-prepared ANTx ceramics, and (f) the fitted Raman shifts for the peaks with wavenumber at $500\text{-}650\text{ cm}^{-2}$. (This figure can be seen as Fig. 5 in the revised manuscript)

Fig. 4 (a) The structure and parameters of Ag1, Ag2 and O ions in AgNbO_3 with Pbcm space group. Rietveld lattice parameters of (b) a , b and c , (c) the (220) and (008) d-spacing, and (d) |B-O| distance as a function of Ta for the as-prepared ANTx ceramics. (This figure can be seen as Supplementary Fig. 6 in the Supplementary Information)

Comment 4: As known, dielectric thin films and nanocomposites have been widely investigated in the field of energy storage recently. Can the authors comment on the advantages of ANTx solid solution for application in energy storage compared with the film and nanocomposite materials?

Response:

Thanks for the valuable suggestion. As demonstrated in this work, the ANTx solid solution exhibits excellent energy storage performance, which make it promising for energy storage application. Generally, bulk ceramics show higher total energy compared with film and nanocomposite, due to the large thickness. Moreover, the preparing process for bulk ceramics is generally simpler without special equipment. The total energy can be further improved if multilayer ceramics are fabricated. It should be admitted that film and nanocomposite generally show higher energy density

than that of bulk ceramics due to the ultrahigh breakdown strength resulting from the significantly decreased sample thickness and the addition of polymer. The good flexibility is another advantage of nanocomposite, which make it promising in flexible electronic device. Recently, film prepared on flexible matrix (mica, for example) was also developed, which opens another way for flexible electronic device application. If ANTx film can be prepared, very good energy density and efficiency can be expected due to its relaxor antiferroelectric feature. Furthermore, the ANTx-PVDF “0-3” nanocomposite may be another promising flexible material system for achieving ultrahigh energy density and efficiency. In this materials system, how to activate the antiferroelectric double-like P - E loops is a challenge and should be addressed.

Based on this suggestion, we added the following outlook paragraph discussing the potentials in the revised paper, also cited recent research on thin film and nanocomposite dielectrics for energy storage and other possible applications.

Outlook: We proposed a strategy of constructing M2-M3 phase boundary with RAFE feature in ANTx solid solution, to simultaneously achieve high energy storage density and efficiency. The energy storage density would be further improved in ANTx multilayer ceramics and film capacitors, due to the significantly increased breakdown strength. In addition, the unfolding of RAFE characteristic of M2-M3 phase boundary on atomic scale in AN-based solid solution gives a solid proof to the long-term confusion on the broad dielectric anomaly over M2-M3 phase transition temperature, which opens a wide range of applications where relaxor feature is desired, such as electrocaloric solid-state cooling devices and hysteresis-free actuators.

Comment 5: The two arrow indicators in Fig. 3b are probably mixed up.

Response:

Thanks for the careful reading and pointing out this mistake. Fig. 3b is updated in the revised manuscript.

Reviewer #4 (Remarks to the Author):

Authors presented an excellent work entitled "Constructing phase boundary in AgNbO₃ antiferroelectrics: pathway simultaneously achieving high energy density and efficiency"

This work provides much awaited breakthrough in high energy density and high efficiency capacitor material for energy storage. This is especially important for futuristic miniaturized devices. The developed material clearly exhibit not only high temperature stability but also high cycle stability. Furthermore, this work provide fundamental understanding of the phenomenon, which could be used as guiding principle for developing such materials for various applications. This paper is suitable for nature communications.

We really appreciate the positive comments and acknowledge the breakthroughs we made in our research.

Here is some suggestions for further improving manuscript:

Comment 1: It will be great if authors could generate a graph showing their results versus other reports on the same topic.

Response:

Thanks for the valuable suggestion. A graph comparing ANTx with other reported dielectrics in the field of dielectric energy storage was added in the revised paper, as shown in Fig. 4d in the revised manuscript. The results reported in this work show great advantage in energy density and efficiency.

Comment 2: Please add some recent and more relevant references. For example:

- (a) Relaxor behavior and electrothermal properties of Sn- and Nb-modified (Ba,Ca)TiO₃ Pb-free ferroelectric.
- (b) A new method for achieving enhanced dielectric response over a wide temperature range.

Response:

Thanks for the good suggestion. Some references related to relaxor behavior and dielectric properties were added in the revised manuscript, to make the research background and discussion of this work more solid and sound.

The cited references are listed as follows.

1. Venkateshwarlu, S., Nayak, S., et al., Relaxor behavior and electrothermal properties of Sn- and Nb-modified (Ba,Ca)TiO₃ Pb-free ferroelectric. *J. Mater. Res.*, 2020, 35(8) :1-11.
2. Maurya, D., Sun, F., et al., A new method for achieving enhanced dielectric response over a wide temperature range. *Sci. Rep.*, 2015, 5(1): 15144-15144.

REVIEWERS' COMMENTS:

Reviewer #1 (Remarks to the Author):

All my previous concerns are cleared in the revised manuscript. I recommend the publication of the manuscript in Nat. comm.

Reviewer #4 (Remarks to the Author):

Authors successfully answered comments and revised manuscript. I don't have further concern.

Reviewers' comments:

Reviewer #1 (Remarks to the Author):

All my previous concerns are cleared in the revised manuscript. I recommend the publication of the manuscript in Nat. comm.

Reviewer #4 (Remarks to the Author):

Authors successfully answered comments and revised manuscript. I don't have further concern.

Point-by-point responses to the reviewers' comments

Reviewer #1 (Remarks to the Author):

Comment:

All my previous concerns are cleared in the revised manuscript. I recommend the publication of the manuscript in Nat. comm.

Response:

Thanks to the reviewer's positive comment. We really appreciate the recommendation to publish the manuscript in Nat. Comm.

Reviewer #4 (Remarks to the Author):

Comment:

Authors successfully answered comments and revised manuscript. I don't have further concern.

Response:

Thanks to the reviewer's positive comment.